# Effects of Quercetin Metabolites on Triglyceride Metabolism of 3T3-L1 Preadipocytes and Mature Adipocytes

**DOI:** 10.3390/ijms20020264

**Published:** 2019-01-11

**Authors:** Itziar Eseberri, Jonatan Miranda, Arrate Lasa, Andrea Mosqueda-Solís, Susana González-Manzano, Celestino Santos-Buelga, Maria P. Portillo

**Affiliations:** 1Nutrition and Obesity Group, Department of Nutrition and Food Science, University of the Basque Country (UPV/EHU) and Lucio Lascaray Research Institute, 01006 Vitoria, Spain; itziar.eseberri@ehu.eus (I.E.); arrate.lasa@ehu.eus (A.L.); andreamosqueda.s@gmail.com (A.M.-S.); mariapuy.portillo@ehu.eus (M.P.P.); 2CIBERobn Physiopathology of Obesity and Nutrition, Institute of Health Carlos III (ISCIII), 28029 Madrid, Spain; 3Grupo de Investigación en Polifenoles (GIP-USAL), Facultad de Farmacia, Universidad de Salamanca, Campus Miguel de Unamuno s/n, 37007 Salamanca, Spain; susanagm@usal.es (S.G.-M.); csb@usal.es (C.S.-B.)

**Keywords:** Quercetin, metabolites, adipocytes, triglycerides

## Abstract

Quercetin (Q) has rapid metabolism, which may make it worthwhile to focus on the potential activity of its metabolites. Our aim was to evaluate the triglyceride-lowering effects of Q metabolites in mature and pre-adipocytes, and to compare them to those induced by Q. 3T3-L1 mature and pre-adipocytes were treated with 0.1, 1 and 10 µM of Q, tamarixetin (TAM), isorhamnetin (ISO), quercetin-3-*O*-glucuronide (3G), quercetin-3-*O*-sulfate (3S), as well as with 3S and quercetin-4-*O*-sulfate (4S) mixture (3S+4S). Triglyceride (TG) content in both cell types, as well as free fatty acid (FFA) and glycerol in the incubation medium of mature adipocytes were measured spectrophotometrically. Gene expression was assessed by RT-PCR. In mature adipocytes, Q decreased TG at 1 and 10 µM, 3S metabolite at 1 and 10 µM, and 3S+4S mixture at 10 µM. 3S treatment modified the glucose uptake, and TG assembling, but not lipolysis or apoptosis. During differentiation, only 10 µM of ISO reduced TG content, as did Q at physiological doses. In conclusion, 3S metabolite but not ISO, 3G, 4S and TAM metabolites can contribute to the in vivo delipidating effect of Q.

## 1. Introduction

Quercetin (Q) is a polyphenol classified as a flavonoid, found, mainly in glycoside form, in a variety of foods including berries, onions and shallots, apples, tea and chocolate [1]. Some of its metabolites, such as Isorhamnetin (ISO) and quercetin-3-*O*-glucuronide (3G), are also present in several food sources [2]. It is estimated that the dietary intake of Q is 5–40 mg/day [3]. However, consumption can reach 200–500 mg/day when fruits and vegetables are abundant in the diet, especially if they are eaten with their skin [4].

Obesity, defined as excess fat accumulation in white adipose tissue, plays a key role as a regulator of lipid storage and release. It can be developed by increasing adipocyte number (hyperplasia) and/or size (hypertrophy) [5]. When hyperplasia takes place, there is a stimulation of pre-adipocyte proliferation and further differentiation. This process, which promotes pre-adipocyte differentiation into mature adipocytes is known as adipogenesis [6]. Nevertheless, this increase in adipocyte number does not necessarily promote obesity directly. Instead, during childhood growth, it determines the lipid-storing capacity of adipose tissue and fat mass in adulthood [7].

For triglyceride synthesis and further storage, mature adipocytes need a source of fatty acids and of glycerol-3-*P*. Fatty acids can be obtained from triglycerides circulating as lipoproteins, due to the action of lipoprotein lipase (LPL), which can be synthesized de novo from Acetyl-CoA or can be taken-up directly from circulation through specific transporters. Glycerol-3-*P* comes from glucose metabolism, after glucose uptake from blood through the glucose transporter GLUT-4. These two molecules are assembled into triglycerides in a process catalyzed by several enzymes. TG stored in adipose tissue can be mobilized in the process known as lipolysis, mediated by three lipases, which implies a breakdown of stored triglycerides and subsequent release of fatty acids and glycerol.

Q has recently been shown to be a potential body fat-lowering molecule. Its positive impact on lipolysis, apoptosis, fatty acid uptake, inhibition of adipogenesis and reduction of lipogenesis has been proposed as its mechanism of action [8,9,10,11,12]. In addition, it seems that its effect on white adipose tissue is accompanied by muscle and liver mitochondrial biogenesis and by improved glycaemic control among other effects, resulting in it being a multi-target flavonoid for body fat reduction [13,14]. Not only abundant cell culture experiments [8,10,15,16], but also animal studies have confirmed its usefulness in body fat reduction, mostly in obese animals [17,18,19,20,21,22]. However, studies in humans remain scarce [19,23,24,25]. A matter of concern in the use of Q as a bioactive molecule is its rapid metabolism, and thus its low bioavailability. Chen et al. [26] determined that 60% of total quercetin ingested by rats was absorbed, and 55.8% of this absorbed amount was metabolized by the gut and 1.8% by the liver. After ingestion, Q is transformed into an aglycone form in the small intestine, that in turn is further metabolized by glucuronidation, sulfatation and methylation reactions [27]. As a result, only a reduced amount of Q and considerable amounts of metabolites reach the bloodstream. According to the literature, the most predominant metabolites in plasma are ISO, tamarixetin (TAM), 3G and quercetin-3-*O*-sulfate (3S) [28,29,30,31] (Figure 1), with glucuronide metabolites being those that appear in higher concentrations and sulfate and methylated those that appear in lower concentrations [32,33,34].

Bearing this in mind, it is not possible to be sure that the fat-lowering properties of Q observed in in vivo experiments are exclusively attributable to Q. The potential activity of its metabolites should not be discarded. Data concerning this issue in adipocytes are scarce so far [35,36,37]. Studies carried out with these molecules in A549 lung cancer cells revealed that Q metabolites could have similar positive effects to those of Q on cell invasion and migration [38].

Considering all these issues, in the present study we wanted to assess whether Q and/or its metabolites are responsible for its beneficial effects in terms of delipidating molecule. For this purpose, the triglyceride-lowering effect of methylated metabolites TAM and ISO, 3G, 3S and a mixture (3S+4S) of 3S and quercetin-4-*O*-sulfate (4S) in pre-adipocytes and mature adipocytes was evaluated and compared to that induced by Q.

## 2. Results and Discussion

In order to address the challenge of determining delipidating capacity of each Q metabolite, several approaches with isolated TAM, ISO, 3G, 3S, and 3S+4S mixture dissolved in ethanol were performed in mature and maturing murine adipocytes. With regard to cell treatment, two aspects must be pointed out. First, the final concentration of ethanol per adipocyte was well below 0.1%, a previously reported non-toxic concentration [39,40,41,42]. Secondly, the assays were conducted with doses lower than those commonly used in cell culture experiments (0.1, 1 and 10 μM). One of the reasons for choosing these doses is related to the prevention of cell integrity, because 10 µM of ISO and 3G was declared as safe in Raw 2647 cells, but toxic effects were observed for both molecules at 25 and 100 µM, respectively [43]. Moreover, in vivo supplementation studies are commonly carried out by using doses of Q that lead to plasma Q and Q metabolite concentrations in the range of nanomolar and micromolar [44,45]. In addition, in previous studies from our laboratory, doses of Q in the range of serum concentrations (≤10 µM) were tested in adipocytes, revealing a dose-dependent effect on triglyceride (TG) reduction in pre-adipocytes [12].

Apart from commercially available metabolites TAM, ISO and 3G, hemisynthetized quercetin sulfate mixture (3S+4S) and 3S were also used for the present study. Appendix A shows the HPLC chromatograms recorded at 370 nm with the two obtained fractions. In the case of the mixture, 4S and 3S represented 33.4% and 62.5% of the recorded peak areas, respectively.

No significant changes in mature adipocyte TG content were observed when these cells were incubated with the lowest dose (0.1 μM) of the molecules studied. At a dose of 1 μM, only the 3S metabolite, among all the molecules tested, reduced TG content. Finally, at 10 µM, Q, 3S+4S and 3S significantly reduced TG content in adipocytes (21%, 20% and 32%, respectively). By contrast, 3G, ISO and TAM were ineffective (Figure 2). Similar results were reported by Lee et al. with ISO treatment in mature adipocytes [35].

In order to explain the TG reduction observed, gene expression of mature adipocyte-specific genes was analyzed at the dose of 10 µM. We chose the highest dose to carry out this analysis because this was the active one for a great number of the molecules analyzed. Treatment with 10 µM of 3S significantly decreased and 3S+4S tended to reduce *lipoprotein lipase* (*lpl*) expression (Figure 3A,B). Evidence confirmed that adipocyte-derived LPL is required for efficient fatty acid uptake and further TG storage in 3T3-L1 adipocytes [46]. It is true that LPL is not determining in in vitro TG accumulation. However, the reduction observed could suggest a positive mechanism of action in in vivo situation. With regard to lipolysis, treatment with 3S+4S tended to reduce *adipose triglyceride lipase* (*atgl*) expression (*p* = 0.09) in mature adipocytes and thus, in order to clarify whether this change could result in changes in this metabolic pathway, glycerol and FFA release were measured (Appendix A). Given that, as previously reported with Q [12], no changes were observed and consequently, it seems that lipolysis is not involved in TG reduction. On the other hand, even though further analysis is needed in order to confirm this fact, it can be proposed that 3S metabolite, alone or in combination with 4S, could act reducing fatty acid uptake.

According to research conducted with adipose tissue explants from lean, overweight, obese and morbidly obese subjects, body fat mass increase is associated with CASP3 and P53 expression elevation and BCL2 expression reduction [47]. Thus, as the apoptotic pathway is related to adipose tissue homeostasis, the potential involvement of 3S metabolite in apoptosis was studied. It promoted remarkably elevated levels of *trp53*, a gene that codifies tumor suppressor p53 protein. While p53 is linked with apoptosis, it has many other roles including cell-cycle arrest, DNA repair or senescence [48]. Due to this fact, other apoptosis-related genes such as *caspase 3* (*cas3*) and the anti-apoptotic gene *bcl2* were assessed (Figure 3A). The expression of both genes revealed apoptosis reduction, instead of promotion with 3S treatment (*bcl2* elevation and *cas3* decrease). Thus, apoptosis does not represent a mechanism of action for 3S metabolite in mature adipocytes. In fact, when 4S was included there was no effect on apoptotic genes. Although the mixture 3S+4S raised the expression of *trp53*, no changes were observed in *cas3* or *bcl2* genes (Figure 3B).

Apart from fatty acid uptake, lipolysis and apoptosis, lipogenesis is another crucial metabolic process involved in fat storage. Uptaken fatty acids or new synthesized ones must be assembled with glycerol in order to accumulate triglyceride inside the adipocyte. As a result, facilitated *glucose transporter member 4* (*glut4*), as well as *diacylglycerol o-acyltransferase* (*dgat*), genes involved in glucose uptake and TG assembly, can be considered limiting genes for TG synthesis. 3S, but not 3S+4S, reduced *glut4*, *dgat1* and *dgat2* gene expression (Figure 3A,B). By contrast, *fatty acid synthase* (*fasn*) related to de novo lipogenesis was not modified by the analyzed molecules. These results suggest that the synthesis of fatty acids is not affected by Q metabolite treatment and TG assembly is reduced.

As far as we know this is the first study to reveal the potential effectiveness of Q metabolites in mature adipocytes, postulating that glucose uptake and TG assembling are mechanisms that could justify the TG reduction observed in mature adipocyte after 3S treatment. Consequently, the effects on body fat observed in animals after Q administration would be due not only to the parent compound but also to this metabolite. It is important to highlight that the addition of 4S metabolite to 3S did not confer any additional effect. In fact, the expression of evaluated genes revealed a decrease in their impact (Figure 3B). These results suggest that the TG-lowering effect can be attributed exclusively to 3S metabolite, and that the addition of 4S results in a dilution of the effective molecule.

It has been described that adipocyte turnover rate for lean and obese humans is around 10% [7]. As mature adipocytes do not have the ability to divide, adipocyte precursors with this capacity must exist in adipose tissue. For this reason, in addition to mature adipocyte analysis, the effects of Q metabolites on pre-adipocytes were also assessed in the present study. ISO has been the most studied of all the Q metabolites in maturing adipocytes. It has been demonstrated that this metabolite reduces TG accumulation, with 10 μM the most effective dose in 3T3-L1 and human adipose tissue-derived stems cells [35,49]. Zhang et al. also reported similar results in 3T3-L1 pre-adipocytes, 12.5 µM being the lowest effective dose [37]. In good accordance with these studies, a significant reduction after 10 µM ISO treatment was observed in maturing pre-adipocytes in the present work. Nevertheless, none of the remaining Q metabolites was able to reduce TG at the doses of 1 or 10 µM, as Q did (Figure 4).

With regard to the mechanisms of action for ISO, Lee et al. [35] demonstrated that nine days of treatment in maturing 3T3-L1 pre-adipocytes with 50 µM reduced adipogenesis through the inhibition of *peroxisome proliferator-activated receptor γ* (*pparγ*) and *CCAAT/enhancer-binding protein α* (*cebpα*). In our cell cultures, the dose of 10 µM tended to decrease *pparγ* gene expression (*p* = 0.06), but not that of *cebpα*, *cebpβ* or *sterol regulatory element-binding factor* (*srebf1*) (Figure 5). By comparing both studies, it is clear that higher doses promote greater impact on adipogenesis inhibition. This conclusion was also reached by Zhang et al. [37], who revealed that ISO treatment decreased adipocyte differentiation at a concentration of 12.5 µM and totally blocked this process at 50 µM. Likewise, they proposed the differentiation stage as a limiting step for ISO effect. According to their data, the inhibitory effect on adipogenesis was less prominent when ISO was added at the latter stages of differentiation. Moreover, they indicated that ISO may control the early differentiation stage by inhibiting the transactivation of pparγ. In the present research, we treated 3T3-L1 pre-adipocytes with ISO throughout the adipogenic process (eight days). Therefore, the weak effect on *pparγ* expression observed could be due not only to the low dose but also to the influence of the differentiating stage.

Much as took place with 3S metabolite treatment of mature adipocytes, the expression of *trp53* was increased after ISO treatment of differentiating pre-adipocytes, but a diminution of mRNA levels of the death repressor *bcl2* was observed (Figure 5). Furthermore, ISO treatment did not promote any change in *cas3* expression. These results reveal that the apoptosis pathway was not completely activated by ISO treatment, probably due to the low dose used. Mirroring mature adipocyte gene analysis, the expression of *dagt1*, *dgat2*, and *glut4* was also evaluated. With the exception of glut4, no changes in the genes mentioned were detected (Figure 5). Therefore, glucose uptake and TG assembly did not justify the observed delipidating effect of ISO.

In spite of the significant effects observed in the expression of several genes the only Q metabolite to induce a significant reduction in pre-adipocyte TG content was ISO, but at 10 μM, a dose higher than that found in serum when animals are treated with Q. Bearing this fact in mind, the involvement of Q metabolites in the anti-obesity effect of this phenolic compound cannot be proposed.

## 3. Materials and Methods

### 3.1. Reagents

Dulbecco’s modified Eagle’s medium (DMEM) was purchased from GIBCO (BRL Life Technologies, Grand Island, NY, USA). Q was purchased from Sigma (St. Louis, MO, USA) and ISO, TAM, and 3G from Extrasynthese (Genay, France). TG were quantified by Infinity Triglycerides reagent (Thermo Electron Corporation, Rockford, IL, USA) and protein concentrations of cell extracts were measured with bicinchoninic acid (BCA) reagent (Thermo Scientific, Rockford, IL, USA). Commercial kits for analyzing FFA and free glycerol were supplied by Roche and Sigma respectively (Free Fatty Acids, Half Micro Test, Roche, Basilea, Sweden and F6428, Sigma, St. Louis, MO, USA).

### 3.2. Synthesis of Quercetin-3-O-Sulfate and Quercetin-4-O-Sulfate Metabolites

#### 3.2.1. Synthesis of Quercetin Sulfates

Quercetin sulfates were synthesized as described by Dueñas et al. [50]. Dry pyridine was added to quercetin (500 mg) to remove possible water associated with quercetin. Pyridine was rotary evaporated, and the dry compound was dissolved in dioxane (50 mL) and allowed to react in a water bath (40 °C) for 90 min with a 10-fold molar excess of sulfur trioxide-*N*-triethylamine complex under nitrogen to avoid contact with air. Products of sulfation precipitated out and stuck to the glass. Dioxane was decanted, and the precipitate was redissolved in 10% methanol in water. The mixtures of quercetin sulfates were fractioned on a Sephadex LH-20 column (350 mm × 30 mm), successively eluted with 10% aqueous ethanol (500 mL) and 20% aqueous ethanol (500 mL). The fractions containing monosulfates were collected, concentrated to dryness under vacuum, redissolved in ultrapure water, and analyzed by high-performance liquid chromatography with diode array and mass spectrometry detection (HPLC-DAD-MS).

#### 3.2.2. HPLC-DAD-MS Analyses

Analyses were carried out with a Hewlett-Packard 1100 chromatograph (Agilent Technologies, Waldbronn, Germany) with a quaternary pump and a DAD coupled to a HP Chem Station (revision A.05.04) data-processing station. Separation was achieved on an Agilent Poroshell 120 EC-C18 column (2.7 µm, 150 mm × 4.6 mm) thermostatted at 35 °C. Solvents used were (A) 0.1% TFA in water and (B) acetonitrile and the elution gradient was from 10 to 15% B for 5 min, from 15 to 25% B for 5 min, from 25 to 35% B over 10 min, from 35 to 50% B over 10 min, isocratic 50% B for 10 min, and re-equilibration of the column, at a flow rate of 0.5 mL/min. Double online detection was carried out in the DAD using 250 and 370 nm as preferred wavelengths and in a mass spectrometer connected to the HPLC system via the DAD cell outlet. MS detection was performed in a Finnigan LCQ detector (Thermoquest, San Jose, CA, USA) equipped with an ESI source and an ion-trap mass analyzer, which were controlled by the LCQ Xcalibur software. Both the auxiliary and sheath gases were nitrogen at flow rates of 20 and 80 L/min, respectively. The source voltage was 4.5 kV; the capillary voltage was 11 V; and the capillary temperature was 220 °C. Spectra were recorded in negative-ion mode between *m*/*z* 150 and 2000. The MS detector was programmed to perform a series of two consecutive scans: a full scan and a MS-MS scan of the most abundant ion in the first scan, using a normalized collision energy of 45%.

#### 3.2.3. Identification and Quantification of Quercetin Sulfates

Chemical hemisynthesis of the quercetin sulfates was performed as described by Dueñas et al. [50]. The complex mixture of products obtained was fractionated on a Sephadex LH-20 column to separate monosulfates from other products (quercetin and quercetin disulfates). Further fractionation by semipreparative HPLC obtained pure quercetin sulfate and a mixture with two quercetin sulfates that were freeze dried for further use.

Chromatograms peaks showed a pseudomolecular ion [M-H]- at *m*/*z* 381 that released a unique fragment at *m*/*z* 301 (−80 amu, loss of a sulfate moiety), corresponding to quercetin. The peaks were identified as quercetin 4’-O-sulfate (peak 1) and quercetin 3’-O-sulfate (peak 2) based on their comparison with compounds previously synthesized and fully identified by NMR [50]. The chromatographic purity of the compounds was calculated to be higher than 96% from the area of the peaks obtained in HPLC chromatograms recorded at 370 and 250 nm.

### 3.3. Experimental Design

3T3-L1 pre-adipocytes, supplied by American Type Culture Collection (Manassas, VA, USA), were cultured in DMEM containing 10% fetal calf serum (FCS). Two days after confluence (day 0), cells were stimulated to differentiate with DMEM containing 10% FCS, 10 µg/mL insulin, 0.5 mM isobutylmethylxanthine, and 1 μM of dexamethasone for two days. From day four onward, the differentiation medium was replaced by FBS/DMEM medium (10%) containing 0.2 µg/mL insulin. This medium was changed every two days until cells were harvested. All media contained 1% Penicillin/Streptomycin (10,000 U/mL), and the media for differentiation and maturation contained 1% (*v*/*v*) of Biotin and Pantothenic Acid. Cells were maintained at 37 °C in a humidified 5% CO2 atmosphere.

### 3.4. Cell Treatment

For the treatment of mature adipocytes, cells grown in 6-well plates were incubated with Q, ISO, TAM, 3G, 3S and 3S+4S at 0.1, 1 and 10 μM (diluted in 95% ethanol) on day 12 after differentiation, because at that day >90% of cells developed mature with visible lipid droplets. In the case of the control group, the same volume of the vehicle (ethanol 95%) was used. Vehicle was diluted 1000-fold in each well, reaching a final concentration of 0.095%. After 24 h, the supernatant was collected and cells were used for TG determination, quantification of glycerol and FFA in the media and RNA extraction. Each experiment was performed three times.

For the treatment of maturing pre-adipocytes, cells grown in 6-well plates were incubated with Q, ISO, TAM, 3G, 3S and 3S+4S at 0.1, 1 and 10 μM (diluted in 95% ethanol) during differentiation. In the case of the control group, the same volume of the vehicle (ethanol 95%) was used. Media containing, or not, molecules were changed every two days: on day 0, day 2, day 4, and day 6. On day 8, the supernatant was collected and cells were used for TG determination, RNA extraction and protein extraction. Each experiment was performed three times.

### 3.5. Measurement of Triacylglycerol Content

After treatment, the medium was removed and cell extracts were used for TG determination. Maturing pre-adipocytes and mature adipocytes were washed extensively with phosphate-buffered saline and incubated three times with 800 µL of hexane/isopropanol (2:1). The total volume was then evaporated by vacuumed centrifugation and the pellet was resuspended in 200 µL Triton X-100 in 1% distilled water. Afterwards, TGs were disrupted by sonication and the content was measured by means of a commercial kit. For protein determinations, cells were lysed in 0.3N NaOH, 0.1% SDS. Protein measurements were performed using the BCA reagent. TG content values were obtained as mg triacylglycerols/mg protein and converted into arbitrary units.

### 3.6. RNA Extraction and RT-PCR

RNA samples from cells treated were extracted using Trizol (Invitrogen, Carlsbad, CA, USA), according to the manufacturer’s instructions. After RNA purity verification, samples were then treated with DNase I kit (Applied Biosystems, Foster city, CA, USA) to remove any contamination with genomic DNA. 1.5 µg of total RNA of each sample was reverse-transcribed to first-strand complementary DNA (cDNA) using iScriptTM cDNA Synthesis Kit (Bio-Rad, Hercules, CA, USA).

Relative *atgl*, *hsl*, *lpl*, *fasn*, *glut4*, *dgat1*, *dgat2*, *bcl2*, *trp53* and *cas3* mRNA levels in mature adipocytes and relative cebpα and cebpβ, srebf1, pparγ, glut4, dgat1, dgat2, *bcl2*, *trp53* and *cas3* mRNA levels in maturing pre-adipocytes were quantified using Real-Time PCR with an iCyclerTM - MyiQTM Real Time PCR Detection System (BioRad, Hercules, CA, USA). For *atgl*, *hsl*, *lpl*, *fasn*, *cebpβ*, *srebf1* and *bcl2* SYBR Green Master Mix (Applied Biosystems, Foster City, CA, USA) was used. The upstream and downstream primers and probe (TibMolbiol, Berlin, Germany, Eurogentec, Liège, Belgium and Metabion, Munich, Germany) are listed in Table 1. *Cebpα*, *pparγ*, *glut4*, *dgat1*, *dgat2*, *trp53*, and *cas3* were measured by TaqMan^®^ Gene Expression Assays (Mm00514283_s1, Mm00440940_m1, Mm01731290_g1, Mm00436615_m1, Mm00515643_m1, Mm499536_m1, Mm01195085_m1 and Mm02619580_g1) and TaqMan^®^ Fast Advanced Master Mix (Applied Biosystems, Foster City, CA, USA). RT-PCR parameters used were those defined by manufacturer’s. β-actin mRNA levels were similarly measured and served as the reference gene.

All gene expression results were expressed as fold changes of threshold cycle (Ct) value relative to controls using the 2^−ΔΔ*C*t^ method [51].

### 3.7. Measurements of Glycerol and Free Fatty Acids in the Media

After treatment in mature adipocytes, aliquots of the medium treated with 10 µM of 3S+4S were removed and analyzed for glycerol and FFA quantification by means of commercial kits (see Reagents paragraph).

### 3.8. Statistical Analysis

Results are presented as mean ± standard error of the mean (SEM). Statistical analysis was performed using SPSS 24.0 (SPSS Inc., Chicago, IL, USA). After confirming the normal distribution of variables using Shapiro-Wilks normality test, each flavonoid dose effect against the control was checked by Student’s t test. Statistical significance was set-up at the *p* < 0.05 level.

## 4. Conclusions

The results obtained in the present study demonstrate that 3S metabolite may contribute to the delipidating effect of Q by reducing glucose uptake and TG assembling in mature adipocytes. ISO metabolite diminished TG accumulation in pre-adipocytes, but at a concentration of 10 μM, which is higher than that found in plasma from animals treated with Q. Consequently, its contribution to the effect of Q should be discarded, as well as that of 3G, 4S and TAM.

## Figures and Tables

**Figure 1 ijms-20-00264-f001:**
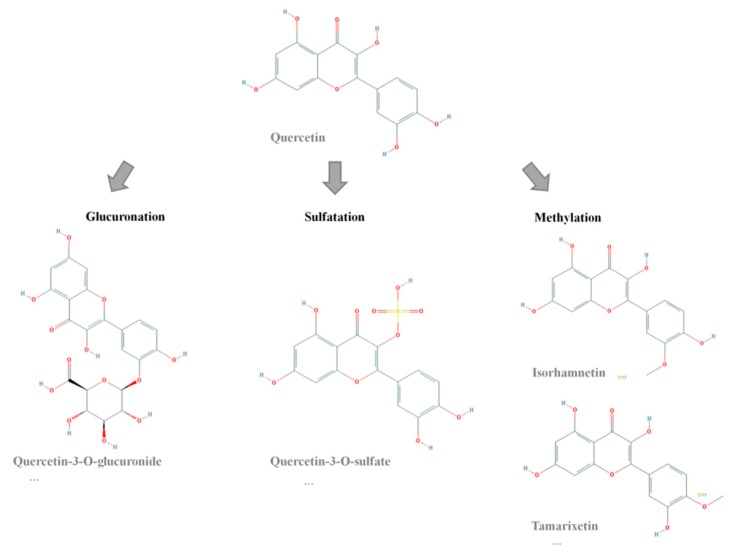
Chemical structures of Q and its metabolites.

**Figure 2 ijms-20-00264-f002:**
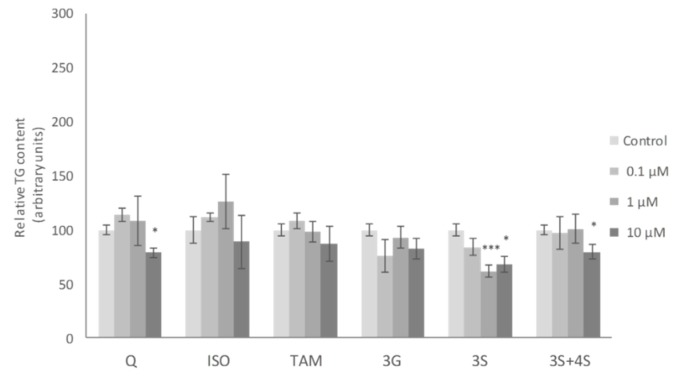
Effects of 0.1, 1 and 10 µM of quercetin (Q), isorhamnetin (ISO), tamarixetin (TAM), quercetin-3-*O*-glucuronide (3G), quercetin-3-O-sulfate (3S), as well as 3S and quercetin-4-*O*-sulfate (4S) mixture (3S+4S) on triacylglycerol content of 3T3-L1 mature adipocytes treated for 24 h. Values are means ± SEM. Comparison between each flavonoid dose and the control was analyzed by Student’s *t*-test. The asterisks represent differences versus the controls (* *p* < 0.05; *** *p* < 0.001).

**Figure 3 ijms-20-00264-f003:**
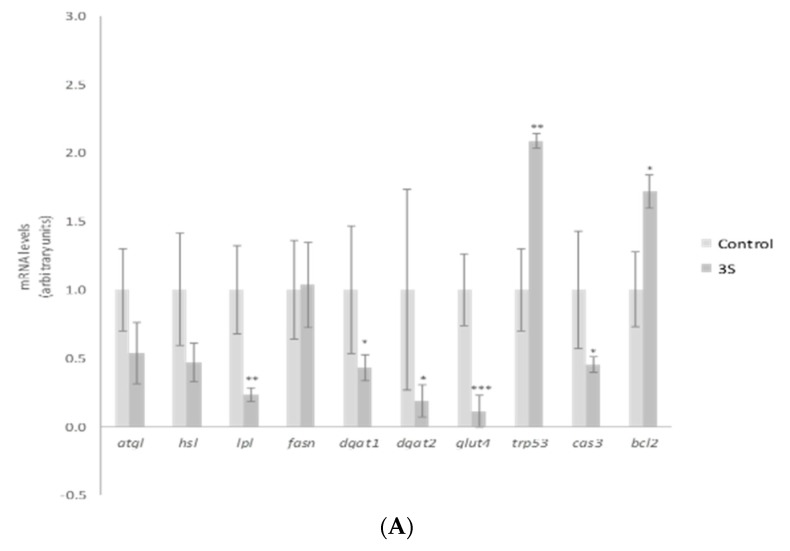
Effects of quercetin-3-*O*-sulfate (3S) (**A**) and quercetin-3-*O*-sulfate and quercetin-4-*O*-sulfate mixture (3S+4S) (**B**) at a dose of 10 µM on the expression of *atgl*, *hsl*, *lpl*, *fasn*, *dgat1*, *dgat2*, *glut4*, *trp53*, *cas3* and *bcl2* in 3T3-L1 mature adipocytes treated for 24 h. Values are means ± SEM. Comparison of 3S or3S+4S and the control for each gene expression was analyzed by Student’s *t*-test. The asterisks represent differences versus the controls (* *p* < 0.05; ** *p* < 0.01; *** *p* < 0.001).

**Figure 4 ijms-20-00264-f004:**
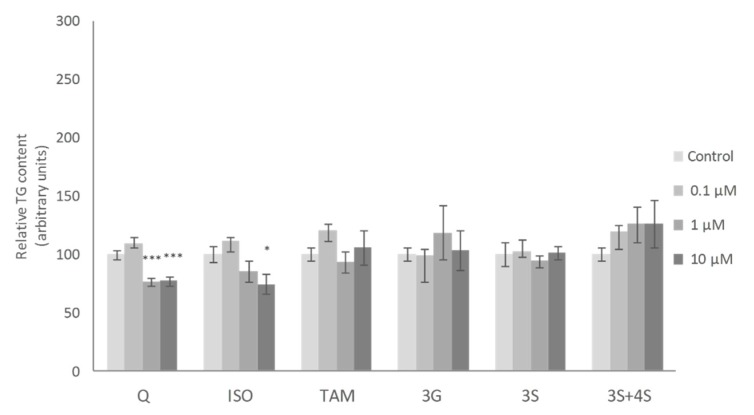
Effects of 0.1, 1 and 10 µM of quercetin (Q), isorhamnetin (ISO), tamarixetin (TAM), quercetin-3-O-glucuronide (3G), quercetin-3-*O*-sulfate (3S), as well as 3S and quercetin-4-*O*-sulfate (4S) mixture (3S+4S) on triacylglycerol content of 3T3-L1 maturing pre-adipocytes treated from day 0 to day 8. Values are means ± SEM. Comparison between each flavonoid dose and the control was analyzed by Student’s *t*-test. The asterisks represent differences versus the controls (* *p* < 0.05; *** *p* < 0.001).

**Figure 5 ijms-20-00264-f005:**
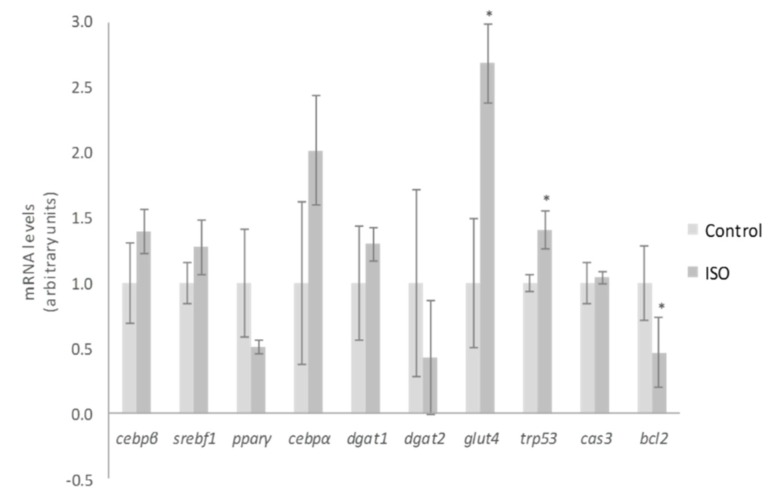
Effects of 10 µM of isorhamnetin (ISO) on the expression of *cebpβ*, *srebf1*, *pparγ*, *cebpα*, *dgat1*, *dgat2*, *glut4*, *trp53*, *cas3* and *bcl2* in 3T3-L1 adipocytes treated for from day 0 to day 8. Values are means ± SEM. Comparison between ISO and the control for each gene expression was analyzed by Student’s *t*-test. The asterisks represent differences versus the control (* *p* < 0.05).

**Table 1 ijms-20-00264-t001:** Primers for PCR amplification of each studied gene.

Gene	Sense Primer	Anti-Sense Primer	Annealing t^a^ (°C)
*atgl*	5′-GAGCTTCGCGTCACCAAC-3′	5′-CACATCTCTCGGAGGACCA-3′	60.0
*hsl*	5′-GGTGACACTCGCAGAAGACAATA-3′	5′-GCCGCCGTGCTGTCTCT-3′	60.0
*lpl*	5′-CAGCTGGGCCTAACTTTGAG-3′	5′-CCTCTCTGCAATCACACGAA-3′	61.5
*fasn*	5′-AGCCCCTCAAGTGCACAGTG-3′	5′-TGCCAATGTGTTTTCCCTGA-3′	60.0
*β-actin*	5′-ACGAGGCCCAGAGCAAGAG-3′	5′ -GGTGTGGTGCCAGATCTTCTC-3′	60.0
*srebf1*	5´- GCTGTTGGCATCCTGCTATC-3′	5′-TAGCTGGAAGTGACGGTGGT-3′	60.0
*cebpβ*	5′-CAAGCTGAGCGACGAGTACA-3′	5′-CAGCTGCTCCACCTTCTTCT-3′	67.5
*bcl2*	5′-AGTACCTGAACCGGCATCTG-3′	5′-GGGGCCATATAGTTCCACAAA-3′	60.0

atgl = adipose triglyceride lipase; hsl = hormone sensitive lipase; lpl = lipoprotein lipase; fasn= fatty acid synthase; srebf1 = sterol regulatory element-binding factor 1; cebpβ = CCAAT-enhancer- binding protein β; bcl2 = B cell leukemia/lymphoma 2.

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
