# Peer review of "Effects of Quercetin Metabolites on Triglyceride Metabolism of 3T3-L1 Preadipocytes and Mature Adipocytes"

_ijms, 2019, doi:10.3390/ijms20020264_

Reviewer 1 Report

In the paper “Effects of quercetin metabolites in 3T3-L1 preadipocytes and mature adipocytes” Eseberri et al. describe potential antilipidemic effects of the major metabolites of quercetin. The choice of tasting the metabolites of quercetin rather than the parent compound in ranges of physiological concentration gave some novelty and interest to the study. However, the study has been poorly designed and the results do not completely support the conclusions.

In general, the manuscript is not well written, the introduction does not provide enough background about obesity and adipose tissue physiology. The “results and discussion” section requires major modifications to clarify why experiments have been performed and to properly discuss the outcome.

However, the major concern regarding the paper is that the results of the experiments do not explain any of the beneficial effects associated with quercetin consumption.

Major comments:

1. Why decreasing lipid content of the adipocyte should be beneficial? One of the main physiological roles of the adipocytes is to store lipids. Similarly, decreasing the differentiation of preadipocytes into adipocyte can be detrimental. Polyphenols are known to exert antioxidant and anti-inflammatory effects. Obesity is related to insulin resistance and chronic inflammation. The authors may consider evaluating the effect on inflammation, secretion of adipokines and insulin sensitivity.

2. Regarding the experiments performed: the paper mainly focused on the lipid content of the adipocytes. Figure 2 shows a reduction in TG following treatment with 3S and 3S+4S, however only asterisks and no bars are present for these two treatments. Is it a problem of the uploaded file?

3. The statistical analyses performed are incorrect (fig 2 and fig 4). Each treatment has been compared to the control using a t-test, the authors should perform an ANOVA, followed by a post hoc test, because many different compounds at different concentrations have been analysed.

4.Following the analysis of glycerol and FFA in the medium and the gene expression pattern, the authors conclude that the reduced TG in the cells is dependent on a reduced FA uptake. However, the results shown are not enough to support this conclusion. First, glycerol and FFA quantifications have been performed only for the association of 3S and 4S, why not 3S alone? To claim an effect on fatty acid uptake the authors should perform an assay for fatty acid uptake that shows a reduction of this process following treatment.

5. How do the authors explain the reduction on GLUT4 expression? Does the treatment cause insulin resistance?

6. Why figure 3 has been divided in two panel? It would have been more correct to have a unique graph showing the two treatments and to analyse it using ANOVA followed by post hoc test.

7. Why the authors decided to evaluate gene related to apoptosis?

8. Cytotoxicity assays need to be performed to ruled out a possible cytotoxic effect of the compounds. The authors cited the treatment of RAW macrophages, but the cytotoxicity might be different in different cell types.

9. How do the authors explain a total lack of effect on TG content after treatment of preadipocyte with 3S and 3S+4S?

10. Line 157-164: It is not clear why the authors decided to evaluate a possible effect on apoptosis. A pro-apoptotic state is associated to insulin resistance, but the dose of ISO was too low to induce this pro-apoptotic state? The entire paragraph should be re-written.

Author Response

In general, the manuscript is not well written, the introduction does not provide enough background about obesity and adipose tissue physiology. The “results and discussion” section requires major modifications to clarify why experiments have been performed and to properly discuss the outcome.

Following the reviewer's suggestion, new paragraphs providing background about obesity and adipose tissue physiology have been added in this revised version (pages 1 and 2, lines 37-53).

In order to clarify why experiments were performed, we have modified the aim in the "Introduction" section. In addition, and following the reviewer recommendation, in this revised version we have started the "Results and Discussion" section by remembering the aim of the study (page 3, lines 86-91), and we have included a new paragraph concerning pre-adipocytes (page 6, lines 214-218).

However, the major concern regarding the paper is that the results of the experiments do not explain any of the beneficial effects associated with quercetin consumption.

As stated by the reviewer, several beneficial effects have been proposed for quercetin apart from those related to white adipose reduction. In response to reviewer´s mayor comments some of them such as the effect of Q and its metabolites on insulin resistance have been addressed. It must be highlighted that the aim of the manuscript was to evaluate the effects of quercetin metabolites on triglyceride metabolism of 3T3-L1 preadipocytes and mature adipocytes. This means that not all the positive effects described for quercetin were assessed. In fact, we only determine the fat-lowering effect of quercetin metabolites in adipocytes, as well as their justificatory mechanism of action according to triglyceride metabolism. In this sense, relevant information can be pointed out from the research. Our in vitro results with isolated quercetin metabolites demonstrated that in vivo body fat-lowering effect attributed to Q, not only corresponds to this flavonoid, but also to the 3S metabolite. Furthermore, fatty acid and glucose uptake and triglyceride assembling in adipocytes were observed as mechanisms of action in triglyceride reduction for the 3S metabolite.

Major comments:

1. Why decreasing lipid content of the adipocyte should be beneficial? One of the main physiological roles of the adipocytes is to store lipids. Similarly, decreasing the differentiation of preadipocytes into adipocyte can be detrimental. Polyphenols are known to exert antioxidant and anti-inflammatory effects. Obesity is related to insulin resistance and chronic inflammation. The authors may consider evaluating the effect on inflammation, secretion of adipokines and insulin sensitivity.

The reviewer is right, delipidating effect on adipocytes and anti-proliferative effect could be deleterious in some situations, because triglycerides can then be accumulated extopically in liver and skeletal muscle, thus leading to lipotoxicity and consequently to insulin resistance. Lipid storage in adipocytes, as well as differentiation of preadipocytes into adipocytes, are physiological processes taking place in order to face excessive energy intake. Due to this fact, if both processes are tackled less amount of defense-systems will show the body against energy imbalance diet. Therefore, the proposed delipidating and anti-proliferative effects must be accompanied with other mechanisms of action. As far as in vivo quercetin effects are concerned, data in the literature show that this phenolic compound reduces body fat accumulation in adipose tissue (similarly to that observed in cultured cells as in the present study), without inducing deleterious effects of liver, muscle and glycaemic control. On the other hand, under some experimental conditions (dose, experimental period length…) quercetin reduces liver fat accumulation, increases hepatic and muscular fatty acid oxidation, improves insulin sensitivity and thus glycaemic control (Synder et al., 2016; Wood dos Santos et al., 2018). Some information about this topic has been included in the revised version of the Introduction section (page 2, lines 56-59).

References:

Snyder et al. J Nutr. 2016 May;146(5):1001-7. doi: 10.3945/jn.115.228817. Epub 2016 Apr 6

Wood dos Santos et al. 2018. Int J Mol Sci. 2018 Sep; 19(9): 2757

With regard to the effect of quercetin metabolites on inflammation, secretion of adipokines and insulin sensitivity, we agree with the reviewer that this would be an interesting approach. However, this was not the purpose of the research. We only wanted to assess the effects of quercetin metabolites on triglyceride metabolism in adipocytes. However and following the reviewer´s comment, the title of the manuscript and the aim of the study were modified in order to adequate them to the conducted research (page 3, lines 81-85).

2. Regarding the experiments performed: the paper mainly focused on the lipid content of the adipocytes. Figure 2 shows a reduction in TG following treatment with 3S and 3S+4S, however only asterisks and no bars are present for these two treatments. Is it a problem of the uploaded file?

We apologize for this mistake but, in fact, there was a problem when uploading the file. Figure 2 has been correctly loaded in the revised version of the manuscript and now asterisks and bars are present.

3. The statistical analyses performed are incorrect (fig 2 and fig 4). Each treatment has been compared to the control using a t-test, the authors should perform an ANOVA, followed by a post hoc test, because many different compounds at different concentrations have been analysed.

ANOVA test is an appropriate statistical test for the comparison among groups. However, in this case, a dichotomic analysis was performed, since the aim of the study was to analyze if each compound was effective or not versus the control group. Thus, each sentence of the discussion has been carefully written and no direct comparison between different compounds has been done.

Although statistical analyses information was provided in the previous version of the manuscript (Material and Methods section and footnotes), in order to avoid any confusion, footnotes, as well as Material and Methods section, have been rewritten for better comprehension. (footnotes and page 9, lines 335-337).

4. Following the analysis of glycerol and FFA in the medium and the gene expression pattern, the authors conclude that the reduced TG in the cells is dependent on a reduced FA uptake. However, the results shown are not enough to support this conclusion. First, glycerol and FFA quantifications have been performed only for the association of 3S and 4S, why not 3S alone? To claim an effect on fatty acid uptake the authors should perform an assay for fatty acid uptake that shows a reduction of this process following treatment.

Authors have carefully revised all data and have realized that even though figures were correct, the tendency to reduce hsl expression mentioned in the previous version was a mistake. In fact, that tendency (P<0.09) was found in atgl gene expression after 3S+4S treatment. Taking into account that 3S did not move lipases expression and 3S+4S tended to reduce that of atgl, authors decided to measure NEFA and glycerol release in the media of 3S+4S treated cells.

We really apologize for this mistake. The discussion section describing these data has been corrected (page 4, lines 123-128 of the revised manuscript). We would like to underline that data were adequately calculated (Figure 3A and 3B were correct) and that the mistake was only in the explanation of the tendency obtained in one of the lipases.

5. How do the authors explain the reduction on GLUT4 expression? Does the treatment cause insulin resistance?

We assessed GLUT4 expression in adipocytes in order to check whether lower glucose uptake is a mechanism responsible for the observed decrease in lipogenesis. According to the obtained results, it is. As stated by the reviewer, it is true that GLUT4 expression reduction is related to insulin resistance. Nevertheless, it is important to point out that glucose uptake from blood-stream takes place mainly in skeletal muscle and not in white adipose tissue. Therefore, in order to confirm whether Q metabolites are causing the insulin resistance GLUT4 expression in myocytes should be analyzed. Unfortunately, this was not the aim of the present research.

6. Why figure 3 has been divided in two panel? It would have been more correct to have a unique graph showing the two treatments and to analyse it using ANOVA followed by post hoc test.

We appreciate the referee´s comment. Nevertheless, as we have answered in the third comment, our goal was to analyze if each compound was effective or not. For this reason, we used student´s t-test analysis and we put the data in two graphics. 

7. Why the authors decided to evaluate gene related to apoptosis?

Genes related to apoptosis were measured because of two reasons. First, because other authors in the literature have observed that body fat mass increase is associated with changes in apoptotic genes, such as CASP3 and P53 elevation and BCL2 reduction (Tinahones et al., 2013; Alkhouri et al., 2010). Therefore, changes in TG content of treated cells in the present study could be associated with changes in this pathway. On the other hand, studies performed with quercetin, and other phenolic compounds, in vitro and in vivo have observed changes in apoptosis and attributed them to the effect of the molecule (Hsu and Yen, 2008).

A sentence explaining why apoptosis was measured has been included in the revised version (page 5, lines 170 and 171).

References:

Tinahones et al. Diabetes Care. 2013 Mar;36(3):513-21. doi: 10.2337/dc12-0194. Epub 2012 Nov 27.

Alkhouri et al. J Biol Chem. 2010 Jan 29;285(5):3428-38. doi: 10.1074/jbc.M109.074252. Epub 2009 Nov 24.

Hsu and Yen. Mol Nutr Food Res. 2008 Jan;52(1):53-61.

8. Cytotoxicity assays need to be performed to ruled out a possible cytotoxic effect of the compounds. The authors cited the treatment of RAW macrophages, but the cytotoxicity might be different in different cell types.

In this experiment viability of cells was not tested. Nevertheless, in other experiments performed in our and other laboratories, the viability of 3T3-L1 cells treated with 25 µM of quercetin has been analyzed and no cytotoxic effect has been observed (Mosqueda-Solis et al., 2017). It is true that other authors have shown that this molecule can exert cytotoxic effects against 3T3-L1 but at 50 µM or higher doses, which are far from the ones used in the present study (Hsu and Yen, 2006). On the other hand, even though there are scarce data about cell viability after quercetin metabolites cell treatment, some authors have recently demonstrated that for example, isorhamnetin-3-O-D-glucuronide does not exert cytotoxic effects at doses of 5, 10, 20, 50 and 100 µM in 3T3-L1 adipocytes (Im et al., 2017). Moreover, and as mentioned in the discussion section, 10 µM of ISO and 3G have been declared as safe in Raw 2647 cells (Boesch-Saadatmandi eta., 2011).

It must be taken into account that, in the present study, when protein content in adipocytes was measured in control and treated cells, no differences were found, meaning that the number of adipocytes remained unchanged. Protein was quantified to express mg TG, per mg of protein. The following table presents obtained protein data after the treatment with the highest dose (10 µM) of each compound:

Differentiation

Protein content (mg/well)

P value

(treatment vs.   control)

Control

0.529 ± 0.013

Quercetin

0.559 ± 0.009

0.11

ISO

0.549 ± 0.018

0.45

TAM

0.510 ± 0.007

0.22

3G

0.558 ± 0.019

0.26

3S

0.574 ± 0.014

0.10

3S+4S

0.519 ± 0.026

0.76

Mature adipocytes

Protein content (mg/well)

P value

(treatment vs.   control)

Control

0.308 ± 0.012

Quercetin

0.278 ± 0.015

0.17

ISO

0.302 ± 0.008

0.83

TAM

0.319 ± 0.008

0.29

3G

0.274 ± 0.017

0.29

3S

0.319 ± 0.002

0.19

3S+4S

0.303 ± 0.029

0.89

Thus, bearing in mind our previous results concerning cell viability in quercetin treatments, data in the literature and the lack of changes in protein content from the present study, authors suggest that there was no cytotoxic effect of the analyzed compounds. 

References:

Mosqueda-Solís et al. Food Funct. 2017 Oct 18;8(10):3576-3586. doi: 10.1039/c7fo00679a.

Hsu et al. Mol Nutr Food Res. 2006 Nov;50(11):1072-9.

Im et al. Pharm Biol. 2017 Dec;55(1):2057-2064. doi: 10.1080/13880209.2017.1357736.

Boesch-Saadatmandi et al. J Nutr Biochem. 2011 Mar;22(3):293-9. doi: 10.1016/j.jnutbio.2010.02.008. Epub 2010 Jun 25.

9. How do the authors explain a total lack of effect on TG content after treatment of preadipocyte with 3S and 3S+4S?

We agree with the referee that it is surprising the lack of effect of sulfate metabolites while there are effective in mature adipocytes. However, it is important to point out that the mechanism of action by which triglyceride reduction occurs in pre-adipocytes and mature adipocytes does not have to be the same. In pre-adipocytes, the treatment is carried out during the differentiation process and thus, bioactive compounds can exert its functions by the regulation of adipogenic genes. By contrast, in the case of mature adipocytes, the treatment is carried out when cells are fully differentiated and thus, the effect is given over other pathways.

Besides, several studies have demonstrated that the biological effect and/or the underlying mechanisms of action of a specific molecule could be different depending on the treatment conditions and the dose. Furthermore, the molecule or its derived metabolites could exert different actions in their target cells. We have observed these facts in our previous studies and also they have been described by other authors.

References:

Lasa et al. Mol Nutr Food Res. 2012 Oct;56(10):1559-68. doi: 10.1002/mnfr.201100772. Epub 2012 Sep 4.

Eseberri et al. Oxid Med Cell Longev. 2015;2015:480943. doi: 10.1155/2015/480943. Epub 2015 Jun 9.

Rayalam et al. Phytother Res. 2008 Oct;22(10):1367-71. doi: 10.1002/ptr.2503.

Suri et al. Biochem Pharmacol. 2008 Sep 1;76(5):645-53. doi: 10.1016/j.bcp.2008.06.010. Epub 2008 Jul 1.

10. Line 157-164: It is not clear why the authors decided to evaluate a possible effect on apoptosis. A pro-apoptotic state is associated to insulin resistance, but the dose of ISO was too low to induce this pro-apoptotic state? The entire paragraph should be re-written.

We agree with the reviewer´s comment that the explanation in the mentioned paragraph is not sufficiently clear. Thus, for a better comprehension of the text, the paragraph has been re-written and relocated (page 4, lines 129-133 and 206-210).

Reviewer 2 Report

The manuscript “Effects of quercetin metabolites in 3T3-L1 preadipocytes and mature adipocytes”. This study is interesting and the findings are novel.  

My major concern

In conclusion, the authors described that “3S metabolite but not ISO , 3G….can contribute to the in vivo delipidating effect of Q.”, because 3S rather than other compounds decreased TG levels in mature adipocytes. However, it can’t rule out the possibility that these metabolites (like 3G) may convert to quercetin aglycone and then exert its bioactivity. In addition the ratio of the metabolites present in the plasma after quercetin intake should be considered.      

Minor issues:

Figure 3 and 5: it should be indicated which bar is the control or the treatment.

Figure 2: the bars of the last two treatment groups are missing.

Author Response

My major concern

In conclusion, the authors described that “3S metabolite but not ISO, 3G….can contribute to the in vivo delipidating effect of Q.”, because 3S rather than other compounds decreased TG levels in mature adipocytes. However, it can’t rule out the possibility that these metabolites (like 3G) may convert to quercetin aglycone and then exert its bioactivity. In addition the ratio of the metabolites present in the plasma after quercetin intake should be considered.     

We acknowledge the referee for this comment. Indeed, it has been postulated that polyphenol-derived metabolites are able to return to the parent compound in target tissues (Walle 2011, Miksits 2009, Maier-Salamon 2006). This conversion happens even more in the case of sulfate metabolites, due to the fact that sulfatases are ubiquitously present in human tissues. Most of the experts in this field believe that further studies are needed about the topic. In this line, one of the approaches could be to determine the bioactive compound. According to the conclusions obtained in the present work, 3S is an effective body fat lowering flavonoid. We consider this information very important because it opens new ideas for future in vivo researches where 3S content in white adipose tissue could be correlated with tissue weight decrease. 

References:

Walle et al. Ann N Y Acad Sci. 2011 Jan;1215:9-15. doi: 10.1111/j.1749-6632.2010.05842.x.

Miksits et al. Planta Med. 2009 Sep;75(11):1227-30. doi: 10.1055/s-0029-1185533. Epub 2009 Apr 6.

Maier-Salamon et al. Pharm Res. 2006 Sep;23(9):2107-15. Epub 2006 Aug 9

As we stated in the Introduction section, quercetin dietary intake is around 5-40 mg/day, but its consumption can reach 200-500mg/day depending on the dietary pattern. It has been analyzed quercetin and derived metabolites plasma concentration after quercetin intake and it has been shown that they can be found at the nanoMolar and microMolar range, the latter especially after quercetin supplementation (Bischoff 2008, Cao 2010). Moreover, and in view of the referee´s interest, the ratio of the metabolites present in the plasma after quercetin intake has been analyzed in the literature. In a study conducted by Justino et al., (2004), after 1 hour of intragastric administration of quercetin in rats, the glucuronide metabolites were the major circulating metabolites and sulfates were the minor ones. Ratio described by these authors was near 8:1 for glucuronide:sulfate metabolites. As they described a part of Quercetin was also methylated to isorhamnetin, but no free isorhamnetin was found and only a small amount of sulfate and glucuronide derivatives of isorhamnetin were found. Human studies have also described that after 1.5 h of quercetin ingestion, quercetin glucosides are not present in plasma and that the major circulating compounds are quercetin glucuronides and in a lower extent, quercetin sulfates (Day et al., 2001). In parallel, it has been described that the methylation process is less important in humans than in rats (Manach et al., 1998), and thus, the presence of this metabolite after quercetin ingestion seems to be very low.

References:

Bischoff et al.  Curr Opin Clin Nutr Metab Care. 2008 Nov;11(6):733-40. doi: 10.1097/MCO.0b013e32831394b8.

Cao et al. Br J Nutr. 2010 Jan;103(2):249-55. doi: 10.1017/S000711450999170X. Epub 2009 Sep 14.

Justino et al. Arch Biochem Biophys. 2004 Dec 1;432(1):109-21.

Day et al. Free Radic Res. 2001 Dec;35(6):941-52.

Manach et al. FEBS Lett. 1998 Apr 24;426(3):331-6.

Minor issues:

Figure 3 and 5: it should be indicated which bar is the control or the treatment.

According to the referee´s comment, figure legends have been included in the revised version.

Figure 2: the bars of the last two treatment groups are missing.

We are sorry for this mistake, there was a problem when uploading the file. Figure 2 has been correctly loaded in the revised version of the manuscript and now asterisks and bars are present. 

Reviewer 3 Report

row 36: tell you " Q showed a potential body fat lowering molecule" and then Q had a positive impact on lipogenesis and adipogenesis....how is this possible? It seems a contradiction

rows 213-215: this phrase is a result

row 219-220: this phrase is a result

rows 234-249: does ethanol not dissolve fats? How is it possible to observe lipid droplets in cells if I use ethanol?

rows 240-241: this phrase is a discussion's data

Author Response

Comments and Suggestions for Authors

row 36: tell you " Q showed a potential body fat lowering molecule" and then Q had a positive impact on lipogenesis and adipogenesis....how is this possible? It seems a contradiction

The referee is right that the sentence seems to be contradictory. The sentence has been rewritten.

rows 213-215: this phrase is a result

The sentence has been moved from Materials and Methods to results and scission section

row 219-220: this phrase is a result

The sentence has been moved from Materials and Methods to results and scission section

rows 234-249: does ethanol not dissolve fats? How is it possible to observe lipid droplets in cells if I use ethanol?

Most used dissolvents for the phenolic compounds treatment in cells are dimethyl sulfoxide (DMSO) and ethanol. As stated by the reviewer, ethanol can dissolve fat and thus lipid droplets. By contrast, DMSO, can induces pores in cell membrane resulting toxic for the adipocyte (Gurtovenko et al 2007). For this reason, in both cases the used concentration is a limiting aspect for the conducted approaches. With regard to ethanol the final concentration in each well was below 0.01%. This dose has been reported a non-toxic concentration, and do not dissolve lipid droplets allowing their visualization (Maeda et al., 2013; Takahashi et al. 2015).  This information has been included in the revised version (page 3, lines 88-90).

References:

Gurtovenko et al. J Phys Chem B. 2007 Sep 6;111(35):10453-60. Epub 2007 Jul 28.

Maeda et al. ISRN Inflamm. 2013 Apr 11;2013:763758. doi: 10.1155/2013/763758. eCollection 2013.

Takahashi et al. FEBS Open Bio. 2015 Jul 2;5:571-8. doi: 10.1016/j.fob.2015.06.012. eCollection 2015.

rows 240-241: this phrase is a discussion's data

The sentence has been included at first paragraph of the Results and discussion section (page 3, lines 88-90).

Round  2

Reviewer 1 Report

The authors addressed most of the comments and the quality of the manuscript improved significantly. However, the data shown are not sufficient to claim an effect on fatty acid uptake. Further analysis should be performed as suggested before.

A minor remark regarding the abstract:

The authors mentioned western blot results that are not presented in the manuscript. This should be modified.

Author Response

RESPONSE TO THE REVIEWER

The authors addressed most of the comments and the quality of the manuscript improved significantly. However, the data shown are not sufficient to claim an effect on fatty acid uptake. Further analysis should be performed as suggested before.
We agree with the reviewer that data in the present study are not enough to conclude that the effect of the 3S metabolite in TG reduction is through a decrease in fatty acid uptake. Therefore, we consider it as a limitation of the study. For this reason, we have modified the sentence concerning these results in the Results section (lines 125-128). Besides, we have modified the final conclusion of the manuscript deleting the sentence where fatty acid uptake reduction was suggested as a possible mechanism of action. This information has been omitted from the abstract and conclusion sections.

A minor remark regarding the abstract: The authors mentioned western blot results that are not presented in the manuscript. This should be modified.

The reviewer is right and we apologize for this mistake. The sentence has been removed from the abstract.

Round  3

Reviewer 1 Report

The authors successfully addressed all the comments.